# Reactivity Studies of Phosphinines: The Selenation of Diphenyl-Phosphine Substituents and Formation of a Chelating Bis(Phosphinine) Palladium(II) Complex

Peter A. Cleaves, Ben Gourlay, Robert J. Newland, Robert Westgate and Stephen M. Mansell *

Institute of Chemical Sciences, School of Engineering and Physical Sciences, Heriot-Watt University, Edinburgh EH14 4AS, UK; petercleaves1@gmail.com (P.A.C.); bg28@hw.ac.uk (B.G.); robnewland@gmail.com (R.J.N.); rw38@hw.ac.uk (R.W.)
* Correspondence: s.mansell@hw.ac.uk; Tel.: +44-131-451-4299

**Abstract:** Phosphinines and donor-substituted phosphinines are of recent interest due to their use in homogeneous catalysis. In this article, a Pd(II) bis(phosphinine) complex was characterised and phosphorus–selenium coupling constants were used to assess the donor properties of the diphenylphosphine substituents of phosphinine ligands to promote their further use in catalysis. The selenation of 2,5-bis(diphenylphosphino)-3,6-dimethylphosphinine (**5**) and 2-diphenylphosphino-3-methyl-6-trimethylsilylphosphinine (**6**) gave the corresponding phosphine selenides **8** and **9**, respectively, leaving the phosphinine ring intact. Multinuclear NMR spectroscopy, mass spectrometry and single crystal X-ray diffraction confirmed the oxidation of all the diphenylphosphine substituents with $^1J_{\text{P-Se}}$ coupling constants determined to be similar to SePPh$_3$, indicating that the phosphinine rings were electronically similar to phenyl substituents. Solutions of **6** were found to react with oxygen slowly to produce the phosphine oxide **10** along with other by-products. The reaction of [bis{3-methyl-6-(trimethylsilyl)phosphinine-2-yl}dimethylsilane] (**4**) with [PdCl$_2$(COD)] gave the chelating dichloropalladium(II) complex, as determined by multinuclear NMR spectroscopy, mass spectrometry and an elemental analysis. The molecular structure of the intermediate **2** in the formation of 4,6-di(*tert*-butyl)-1,3,2-diazaphosphinine (**3**) was also determined, which confirmed the structure of the diazaphosphacycle P(Cl){N=C(*t*Bu)CH=C(*t*Bu)-N(H)}.

**Keywords:** phosphinine; phosphorus ligands; palladium; selenium-phosphorus coupling constants; ligand properties

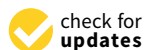



## 1. Introduction

Phosphinines are the phosphorus analogue of pyridine. The initial discovery of 2,4,6-triphenylphosphinine by Märkl [1] and the parent, unsubstituted phosphinine by Ashe [2] are now landmarks in low-coordinate main group chemistry and continue to inspire new research in the field [3,4]. Recent developments have focused on the use of phosphinines as ligands [5–10], particularly in homogeneous catalysis [11–15] where they show interesting differences to conventional phosphine ligands. Examples from our group include the use of phosphinophosphinine complexes as small bite-angle ligands [16] for Ru-catalysed transfer hydrogenation, alcohol upgrading and acceptorless dehydrogenation reactions [17,18] as well as Cr catalysts for ethylene oligomerisation reactions producing unusual alkyl- and alkenylcyclopentane products [19] and Rh complexes for the hydroboration of carbonyls [20]. Others researchers have demonstrated the exceptional behaviour of Rh-phosphinine complexes in hydroformylation [21,22] and hydrogenation reactions [23], Au-phosphinine catalysts for cycloisomerisations [24] and cyclometalated iridium(III)–phosphinine precatalysts for water oxidation reactions [25] among many others [5,11].

The oxidation of phosphines with sulphur and selenium is well-known [26,27] with phosphine selenides being particularly useful in determining the electronic properties

of the parent phosphines via the measurement of phosphorus–selenium coupling constants [28–30]. With respect to phosphinine chemistry, oxidation with sulfur and selenium is usually difficult with phosphinine sulfides initially only tentatively characterised as intermediates in the reaction of S with 2-Ph-4,5-Me$_2$PC$_5$H$_2$ (via trapping reactions with 2,3-dimethylbutadiene) and S with 2,3-(PPh$_2$)$_2$-6-PhPC$_5$H (ca. 20% yield before a continued reaction led to decomposition), principally identified by $^{31}$P NMR spectroscopy [31,32]. The analogous reaction with Se gave only weak evidence of a phosphinine selenide [31]. The first well-characterised phosphinine sulfides were formed from the reactions of 2,6-R$_2$-3,5-Ph$_2$PC$_5$H (R = Ph, SiMe$_3$) with S in toluene at 90 °C for 5–7 days and revealed $^{31}$P chemical shifts (159 and 194 ppm, respectively) greatly reduced from the starting materials (206 and 269 ppm, respectively) [33]. Their molecular structures were determined by X-ray crystallography; P=S bond lengths (1.916(1) and 1.929(1) Å, respectively) shorter than those in S=PPh$_3$ (1.952 Å) were observed [33]. Additional phosphinine sulfides were observed as products in the reaction of S with either a pyridyl-substituted phosphinine or with 2,4,6-triphenylphosphinine in the presence of pyridine [34]. The first well-characterised and quantitatively formed phosphinine selenide was only synthesised recently using red selenium and 2,6-(SiMe$_3$)$_2$PC$_5$H$_3$; a $^{31}$P chemical shift of 170 ppm was observed [35].

Phosphinophosphinines are useful ligands for transition metal complexes [5,9] and form four-membered chelates that can be extended into five-membered chelates through oxidation of the phosphine with sulfur, which has been well-studied [10], or oxygen, selenium and organic azides [36], which are almost unknown. For example, phosphinines bearing two ortho-P(S)Ph$_2$ groups were cleanly synthesised from analogous diphenylphosphine-substituted phosphinines by a reaction with elemental sulfur [37,38]. These species were utilised as ligands in their phosphahexadienyl anion form after an attack on the phosphorus by suitable nucleophiles, e.g., lithium alkyls, chloride and alkoxide [39–44]. Phosphinines with one phosphine-sulfide substituent have also been accessed via the reaction of an azaphosphinine with a phosphine-sulfide-substituted alkyne [45]. In this contribution, we describe the reaction of phosphinophosphinines with selenium to better characterise the donor properties of the PPh$_2$ substituents, we establish that the phosphinine moiety is left untouched under these conditions and we compare the reactivity of phosphinophosphinines with oxygen from the air. We also describe the synthesis of a bis(phosphinine) palladium complex and crystallographic characterisation of the chlorodiazaphosphacyclic intermediate formed in the synthesis of the key starting material, 1,3,2-diazaphosphinine.

## 2. Results

The phosphinines in this work were all synthesised using the 1,3,2-diazaphosphinine methodology pioneered by Le Floch, Mathey and co-workers [46,47]. PCl$_3$ was reacted with a titanocycle (**1**) [48] formed from the reaction of Cp$_2$TiMe$_2$ with two equivalents of NC$^t$Bu [49] followed by the addition of NEt$_3$ (Scheme 1) [46,47]. The anticipated intermediate in this synthesis was a chlorodiazaphosphacycle (**2**). We were fortunate to crystallise this air- and moisture-sensitive compound in the course of attempting to isolate **3**. NMR spectroscopy revealed a $^{31}$P resonance at 109 ppm and the same resonance was also observed upon the addition of HCl in ether to 1,3,2-diazaphosphinine (**3**).

Single crystal X-ray diffraction experiments revealed the molecular structure of **2** (Figure 1). The phosphorus atom was situated in an almost planar six-membered ring with single bonds to two nitrogen atoms (1.689(3) and 1.653(3) Å to N1 and N2, respectively). There was evidence of alternating single (C2-C3 1.432(4) Å) and double bonds (e.g., C1-C2 = 1.359(4) Å) around the ring albeit with the C-C single bond shorter than usual. The Cl atom was situated perpendicular to the plane of the ring and there was a stereochemically active lone pair on the phosphorus. The location of the double bonds was the same as in the parent titanocycle (**1**), with an N-H bond present rather than a methylene unit; the addition of HCl to **3** formed P-Cl and N-H bonds, as was expected from the higher electronegativity of nitrogen compared with phosphorus. Six-membered heterocycles featuring an N-P-N linkage are not uncommon in the Cambridge Structural

Database but there are only a handful of examples with a P-Cl substituent and all feature an exocyclic substituent on both N atoms [50–55]. Other related examples are cationic and supported by a β-diketiminate framework [56,57].

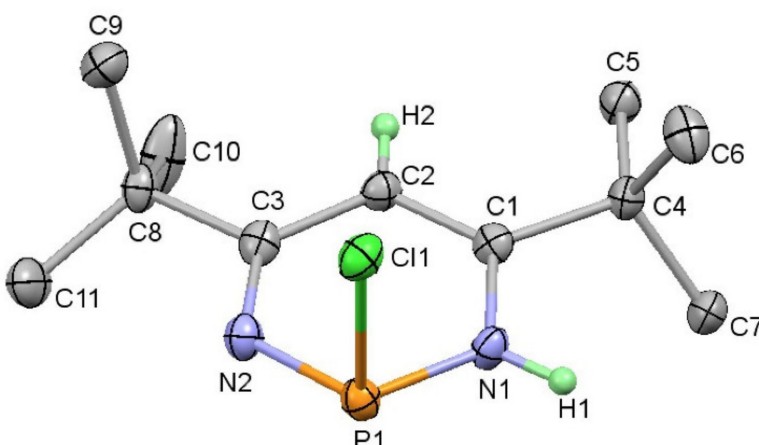

**Scheme 1.** Formation and reactions of phosphinines.

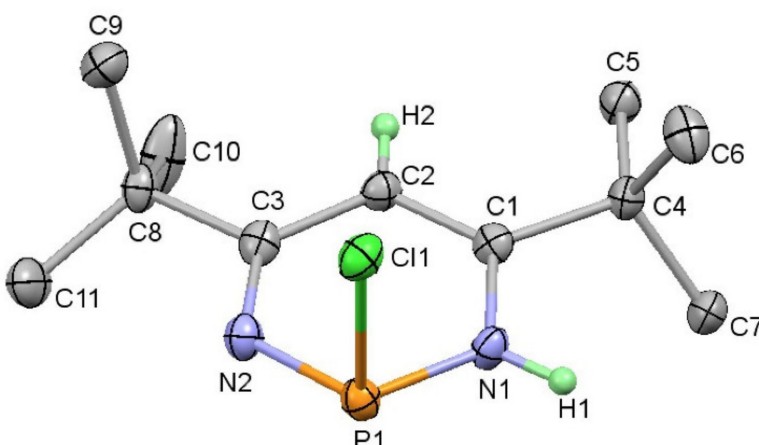

**Figure 1.** Molecular structure of **2** (thermal ellipsoids at 50%; all H atoms except for those attached to the heterocyclic ring have been omitted for clarity). Selected bond lengths (Å) and angles (°): Cl1-P1 2.2417(11); P1-N1 1.689(3); P1-N2 1.653(3); N1-C1 1.377(4); C1-C2 1.359(4); C2-C3 1.432(4); N2-C3 1.320(4); N1-P1-Cl1 99.59(10); N2-P1-Cl1 98.29(10); N2-P1-N1 101.55(13).

The phosphinines **4** [58], **5** [36] and **6** [18] all featured additional P donors, either another phosphinine ring (**4**) or PPh$_2$ moieties. They were synthesised by the reaction of diazaphosphinine (**3**) with alkynes in two sequential steps due to the slower reaction of the intermediate azaphosphinines with alkynes. This allowed two different alkynes to be used in the formation of **4** and **6**. A reaction of the bis(phosphinine) (**4**) with [PdCl$_2$(COD)]

was attempted to ascertain whether the cyclooctadiene ligand (COD) could be displaced by the bis(phosphinine) and whether a chelated bis(phosphinine) complex was preferred over two monodentate ligands. The reaction of [PdCl$_2$(COD)] with two equivalents of **4** led to the chelate complex **7** in preference to [PdCl$_2$(**4**)$_2$]; matching the stoichiometry 1:1 gave **7** in a 51% yield as a poorly soluble red solid. The reaction was selective with only one product observed. Unfortunately, we were unsuccessful in growing single crystals suitable for X-ray diffraction. $^{31}$P{$^1$H} NMR spectroscopy showed a singlet at 204 ppm, which was at a lower frequency when compared with **4** (261 ppm). This shift to a lower frequency upon coordination was not seen for [RuCp*(Cl)(**4**)] or [RuCp*(H)(**4**)], which gave higher frequency resonances than the free ligand (273 and 286 ppm, respectively) although the [Mo(CO)$_4$(**6**)] and [W(CO)$_4$(**6**)] complexes gave lower $^{31}$P chemical shifts for the coordinated phosphinine ligand compared with **6**. $^1$H NMR spectroscopy revealed 2 multiplets for the H atoms at the 4- and 5-positions along with 1 resonance for the 3-Me group and 1 SiMe$_3$ environment in agreement with a plane of symmetry passing through the palladium and the SiMe$_2$ group. However, the presence of two resonances for the SiMe$_2$ group indicated that the ligand was not coplanar with the PdCl$_2$ fragment, as has been seen previously in complexes of this ligand with other metal fragments [58]. We attributed this flexibility to bind to metals out of the plane of the ligand to the high s-character of the phosphinine lone pairs, which render them less directional [11]. Mass spectrometry revealed the correct isotope distribution and accurate mass for the [M-Cl]$^+$ ion and an elemental analysis was also in agreement with the empirical formula of **7**. Pd-phosphinine complexes are relatively rare. Monodentate complexes of 2,4,6-tri-*tert*-butylphosphinine with Pd(II), Pt(II) and Au(I) have been reported [59] along with chelating complexes of 2-pyridyl-4,6-diphenylphosphinine to PdCl$_2$, which was poorly soluble and featured a $^{31}$P NMR chemical shift of 159 ppm [60], as well as [PdCl$_2${2-(PPh$_2$O)PC$_5$H$_4$}], which gave a doublet at 165.9 ppm for the phosphinine P atom [61]. Other Pd complexes usually feature P-substituted phosphahexadienyl anions [38–40,45,62]. Interestingly, the phosphine-sulfide-substituted phosphinine 1,2-{P(S)Ph$_2$}$_2$-3,5-Ph$_2$PC$_5$H reacted with [PdCl$_2$(COD)] to give a phosphahexadienyl ligand featuring a P-Cl bond driven by the coordination of the sulfur atoms and the stability of the d$^8$ square planar geometry [37].

Reactions with selenium were attempted to investigate the selenation of the diphenylphosphine groups and to establish the inertness of the phosphinine to oxidation under these conditions. Reactions of **5** with excess selenium gave a mixture of products with short reaction times, which was likely to include the monoselenated products of the PPh$_2$ groups as $^{31}$P NMR spectroscopy revealed new phosphinine $^{31}$P doublets at 239.4 and 231.9 ppm (103.4 Hz and 100.8 Hz, respectively). For comparison, the phosphinine $^{31}$P NMR signal for **5** was a doublet (29 Hz) at 220.8 ppm. Continued reaction gave full conversion to the diselenated product (**8**) with $^{31}$P NMR resonances evident at $\delta$ = 234.8 (d, 100 Hz, phosphinine), 32.9 (dd, 100 and 6 Hz, ortho-PPh$_2$) and 32.4 ppm (d, 6 Hz, meta-PPh$_2$) with selenium satellites only present on the lower frequency resonances (728 Hz for the ortho-PPh$_2$ and 734 Hz for the meta-PPh$_2$). The $^{77}$Se NMR spectrum showed the expected two doublets with the same coupling constants, allowing the assignment of these signals. A comparison with SePPh$_3$, which has a $^1J$ coupling constant of 732 Hz [29], showed that there was very little difference in the donor ability so the phosphinine substituent acted similarly to a phenyl ring. The characterisation of **8** was performed by mass spectrometry (accurate mass determined for [M + H]$^+$) and by single crystal X-ray diffraction. The molecular structure (Figure 2) revealed disorder in the central phosphinine ring over an inversion centre situated in the middle of the molecule, as was seen in the structure of **5** [36], which prevented a detailed assessment of the bond lengths and angles in the ring. The P=Se bond lengths were determined to be 2.1097(5) Å.

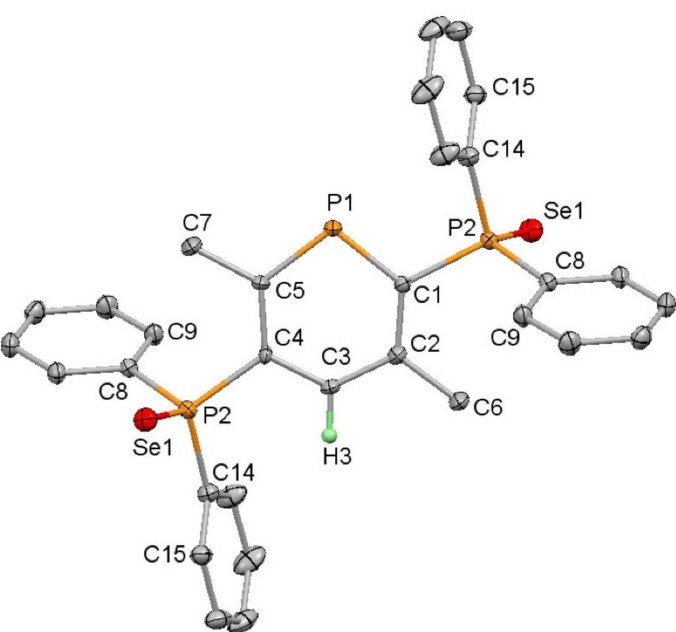

**Figure 2.** Molecular structure of **8** (thermal ellipsoids at 50%; all H atoms except for those attached to the heterocyclic ring have been omitted for clarity). Only one position of the disordered phosphinine ring is shown.

Reactions of phosphinophosphinine (**6**) with grey Se gave the analogous phosphine selenide (**9**) with no evidence of the formation of a phosphinine-selenide despite extensive heating (100 °C for 1 week) and excess Se (five equivalents). $^{31}P\{^1H\}$ NMR spectroscopic resonances for **9** were observed as doublets at 249.0 and 34.4 ppm (100.6 Hz) with selenium satellites present on the lower frequency resonance (749 Hz). This indicated a marginally poorer σ-donor ability than for PPh$_3$ (732 Hz) and identical behaviour to diphenyl-2-pyridylphosphine (749 Hz) [30]. Additional characterisation was achieved by $^1H$ NMR spectroscopy as well as accurate mass spectrometry and an elemental analysis. Single crystal X-ray diffraction revealed the anticipated molecular structure (Figure 3). The phosphinine ring was planar with C-C bond lengths similar to those observed in other aromatic phosphinine rings [18,19,36] and benzene. The P-C bond lengths in the ring (1.7516(17) and 1.7282(18) Å) were shorter than the P-C bond length to the PPh$_2$ substituent (1.8235(17) Å). The Se=P bond length was 2.1132(4) Å, indistinguishable within the error to Se=P in **8**. The structure of 2-{P(S)Ph$_2$}-3-Me-5,6-Ph$_2$PC$_5$H was similar and featured typical bond lengths and angles for an aromatic phosphinine heterocycle [45].

We were intrigued by the differences between bis(phosphino)phosphinines and silylphosphinines [63–65] with respect to their air and moisture stability. We found the doubly silyl-substituted bis(phosphinine) (**4**) to be unstable under atmospheric conditions, requiring column chromatography under anaerobic conditions as well as storage and handling under N$_2$. The bis(phosphino)phosphinine (**5**); on the other hand, was stable to air and moisture and this compound could be extracted into hot hexane by Soxhlet extraction over many days with no precautions required to exclude air or moisture. We found that the stability of silyl-substituted phosphinophosphinine (**6**) in the solution was intermediate between these two extremes. Monitoring an aerobic CDCl$_3$ solution of **6** by $^{31}P\{^1H\}$ NMR spectroscopy revealed that after 21 days at room temperature, two additional species were present along with **6**. We assigned one product as the phosphine oxide (**10**) based on: (i) the $^{31}P$ chemical shifts of the two doublets at 256.2 and 32.1 ppm that were in the correct regions for phosphinine and phosphine oxide; (ii) the $^2J$ coupling of 103 Hz, which was similar to the analogous selenide (**6**) (101 Hz); and (iii) an accurate mass spectrum identifying a species with $m/z$ = 383.1144 for [**10** + H]$^+$. The other product (**11**) was evident as two doublets in the $^{31}P\{^1H\}$ NMR spectrum at −3.0 and −14.0 ppm with a 31 Hz coupling. Rough estimations by a $^{31}P$ NMR integration

revealed ca. 10% **10**, 70% **6** and 20% **11**. Heating to 40 °C for 21 days revealed a continuing reaction but a further 21 days of heating showed that **11** was not stable; only resonances for **10** and decomposition were evident. Due to the lack of a phosphinine resonance, we suspected that **11** resulted from an attack at a P=C double bond, probably by water with further reactivity a possibility including the loss of the SiMe$_3$ group and/or cycloadditions, as has been seen previously [19,36].

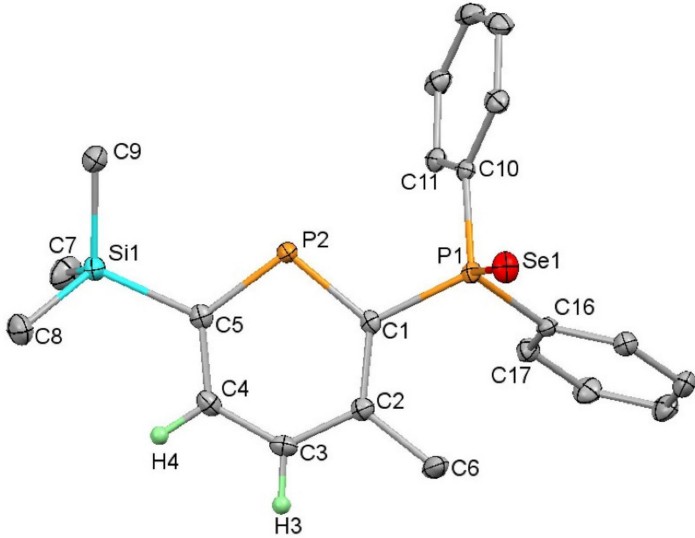

**Figure 3.** Molecular structure of **9** (thermal ellipsoids at 50%; all H atoms except for those attached to the heterocyclic ring have been omitted for clarity). Selected bond lengths (Å) and angles (°): Se1-P1 2.1132(4); P2-C1 1.7516(17); C1-C2 1.407(2); C2-C3 1.397(2); C3-C4 1.386(2); C4-C5 1.403(2); P2-C5 1.7282(18); P1-C1 1.8235(17); C5-P2-C1 103.30(8).

## 3. Conclusions

The structure of the intermediate chlorodiazaphosphacycle (**2**) in the synthesis of 1,3,2-diazaphosphinine was confirmed. It revealed that its structure was analogous to the parent titanacycle (**1**) with the same pattern of single and double bonds around the ring, P(Cl){N=C($^t$Bu)CH=C($^t$Bu)-N(H)}. A SiMe$_2$-linked bis(phosphinine) (**4**) reacted with [PdCl$_2$(COD)] to form the chelating complex and was characterised thoroughly despite its poor solubility. However, single crystals have not been achieved (to date), precluding a structural characterisation by single crystal X-ray diffraction. The reactions of grey selenium with phosphinophosphinines generated the phosphine selenide derivatives whilst leaving the phosphinine ring intact. This was as anticipated as phosphinine selenides are extremely rare. The $^1J_{P-Se}$ coupling constants showed that the phosphinine ring acted similarly to a phenyl ring. This was in line with the favourable binding to transition metals already established as 2-phosphinophosphinines have been shown to act as competent ligands for a variety of transition metal fragments. Chloroform solutions of 2-PPh$_2$-3-Me-6-SiMe$_3$PC$_5$H$_2$ were shown to undergo a slow reaction with oxygen over the course of several weeks to form the phosphine oxide-substituted phosphinine. Side reactions also occurred and $^{31}$P NMR spectroscopy showed that the phosphinine ring did not remain intact in these reactions.

## 4. Materials and Methods

All reactions requiring inert conditions were performed under an oxygen-free nitrogen atmosphere by using standard Schlenk line techniques or by using an MBRUAN UNILab Plus glovebox, unless otherwise noted. Dry toluene was obtained from a solvent purification system (MBRAUN SP-300) and stored over 4 Å molecular sieves prior to use. When required to be dry, C$_6$D$_6$ and CDCl$_3$ were dried over activated 4 Å molecular sieves prior to use. Non-dry solvents were used as received from Fisher Scientific or Goss Scientific (deuterated solvents). Cp$_2$TiMe$_2$ [66], diazaphosphinine [46,47] and the phosphinines **4** [58],

**5** [36] and **6** [18] were synthesised as previously described. NMR spectra were obtained using either a Bruker AVIII400 (400 MHz) or an AVIIHD (400 MHz) spectrometer. $^1$H NMR spectra were recorded at 400 MHz and referenced to the residual protio solvent peak (7.24 for CHCl$_3$ in CDCl$_3$ and 7.16 for C$_6$D$_5$H in C$_6$D$_6$). The $^{13}$C{$^1$H} NMR spectra were recorded at 101 MHz and referenced to the solvent peak (77.23 ppm for CDCl$_3$ and 128.39 ppm for C$_6$D$_6$). $^{31}$P{$^1$H} NMR spectra were recorded at 162 MHz and referenced to an external standard. Mass spectrometry was conducted at the National Mass Spectrometry Facility at Swansea University using the techniques stated. Elemental analyses were performed by Dr. Brian Hutton (Heriot-Watt University).

Single crystals suitable for X-ray diffraction were covered in inert oil and placed under the cold stream of a Bruker D8 Venture or a Bruker X8 APEXII four circle diffractometer cooled to 100 K. The exposures were collected using Mo K$\alpha$ radiation ($\lambda$ = 0.71073). Indexing, data collection and absorption corrections were performed. The structures were then solved using SHELXT [67] and refined by a full-matrix least-squares refinement (SHELXL) [68] interfaced with the programme OLEX2 [69]; **2** was refined as a two component twin (0.4564(18):0.5436(18)) and **8** showed a disorder of the phosphinine ring and a toluene solvate molecule over two positions that was modelled successfully.

### 4.1. Preparation of Chlorodiazaphosphacycle (*2*)

Method a: PCl$_3$ and NEt$_3$ were added to titanacycle (**1**) according to the standard literature method [46,47]. Toluene was removed from the solution under a reduced pressure and the residue was distilled under a reduced pressure into a liquid nitrogen-cooled trap. A few single crystals were observed along with an oil.

Method b: HCl in diethyl ether was added to a solution of diazaphosphinine in toluene. Upon cooling in the freezer, a small amount of precipitate formed, which was isolated by filtration and then analysed.

**$^{31}$P{$^1$H} (162 MHz, CDCl$_3$, 298 K)** $\delta$ = 109.1 ppm; **$^1$H NMR (400 MHz, CDCl$_3$, 298 K)** $\delta$ = 6.25 (1 H, d, $^4$J$_{H-P}$ = 1.5 Hz, CH), 1.26 (18 H, s, $^t$Bu).

### 4.2. Preparation of [PdCl$_2$($\kappa$:$\eta^1$:$\eta^1$-**4**)] (*7*)

Toluene (50 cm$^3$) was added to a mixture of [PdCl$_2$(COD)] (0.150 g, 0.53 mmol) and bis(phosphinine) (**4**) (0.526 g, 1.25 mmol) at room temperature. The red-orange suspension was heated for 3 h at 70 °C then allowed to cool to room temperature and settle over 16 h to afford a red solid and a red solution. The solution was separated from the solid and NMR spectroscopic studies of the solid (dissolved in CDCl$_3$) revealed a singlet $^{31}$P{$^1$H} resonance at 204 ppm, indicative of chelating $\eta^1$:$\eta^1$ coordination. The red solution, which contained bis(phosphinine), was then reacted with a further equivalent of [PdCl$_2$(COD)] (0.150 g, 0.53 mmol) at 70 °C for 3 h, which produced a further red solid. The red solids were then combined (0.266 g, 0.54 mmol, 51%). Unfortunately, crystals suitable for X-ray diffraction studies could not be grown.

**$^{31}$P{$^1$H} (162 MHz, CDCl$_3$, 298 K)** $\delta$ = 203.9 ppm; **$^1$H NMR (400 MHz, CDCl$_3$, 298 K)** $\delta$ = 7.94 (m, 2H, meta-Ar-*H*), 7.18 (m, 2H, para-Ar-*H*), 2.63 (s, 6H, Ar-*Me*), 1.54 (s, 3H, SiMe), 1.12 (s, 3H, SiMe), 0.57 ppm (s, 18H, SiMe$_3$). Anal. calcd for C$_{20}$H$_{34}$Si$_3$P$_2$Cl$_2$Pd: C 40.17; H 5.73; N 0. Found: C 40.34; H 5.62; N 0. HRMS (ASAP/TOF) m/z: calcd for C$_{20}$H$_{34}$Si$_3$P$_2$Cl$^{102}$Pd: 557.0188[M-Cl]$^+$, found: 557.0192.

### 4.3. Preparation of 2,5-Bis(diphenylphosphineselenide)-3,6-dimethylphosphinine (*8*)

Under nitrogen, **5** (198 mg, 0.40 mmol), Se (104 mg, 1.31 mmol) and toluene (2 cm$^3$) were sealed in an ampoule equipped with a J. Young tap and heated at 110 °C for 24 h. The reaction mixture was filtered to remove the excess selenium and all volatiles were removed under a reduced pressure. THF (10 cm$^3$) was added to the solid and heated to 50 °C to dissolve the product and the solution was filtered. The THF was then removed under a reduced pressure, yielding a yellow powder (120 mg, 0.18 mmol, 45%).

**$^1$H NMR (400 MHz, CDCl$_3$, 298 K)** $\delta$ = 7.84 (dd, 13.6 and 7.7 Hz, 8H, meta-Ar), 7.51-7.44 (m, 12H, ortho- and para-Ar), 6.90 (dt, 1H, $^3J_{H-P}$ = 16.1 Hz, $^4J_{H-P}$ = 3.4 Hz, H$_C$), 2.58 (d, 3H, $^3J_{H-P}$ = 17.9 Hz, H$_F$), 2.30 (s, 3H, H$_G$); **$^{31}$P{$^1$H} NMR (162 MHz, CDCl$_3$, 298 K)** $\delta$ = 234.8 (d, $^2J_{P1-P2}$ = 100 Hz, P$_1$), 32.9 (dd, $^2J_{P1-P2}$ = 100 Hz, $^5J_{P2-P3}$ = 6 Hz, $^1J_{P-Se}$ = 728 Hz, P$_2$), 32.4 (d, $^5J_{P2-P3}$ = 6 Hz, $^1J_{P-Se}$ = 734 Hz, P$_3$). **$^{77}$Se NMR (76 Hz, CDCl$_3$, 298 K)** $\delta$ = –259.4 (d, 728 Hz, Se=P$_2$), –286.3 (d, 734 Hz, Se=P$_3$); **$^{13}$C{$^1$H}(100.6 MHz, CDCl$_3$, 298 K)** $\delta$ = 168.6 (d), 167.9 (dd), 147.2 (m), 139.1 (m), 133.4 (dd), 132.9 (d), 132.2 (dd), 131.4 (d), 130.5 (dd), 130.2 (d), 128.9 (dd), 25.3 (d, C$_G$), 24.1(dd, C$_F$); **HRMS (ASAP/TOF):** calcd. for C$_{31}$H$_{28}$P$_3$$^{74}$Se$^{76}$Se: 642.9821 [M+H]$^+$, found: 642.9837 m/z; calcd. for C$_{31}$H$_{28}$P$_3$Se$^+$: 573.0564 [M + H − Se]$^+$, found: 573.0581.

*4.4. 2-Diphenylphosphineselenide-3-methyl-6-trimethylsilylphosphinine (**9**)*

An ampoule was charged with **6** (100 mg, 0.27 mmol), selenium powder (50 mg, 0.63 mmol, 2.3 equiv.) and toluene (2 cm$^3$) then sealed with a J. Young tap. The reaction was heated to 120 °C for 18 h then filtered to remove the excess selenium. The filtrate was concentrated to half its volume and stored at −25 °C for one week. Filtration followed by drying under a high vacuum yielded **9** (80 mg, 0.18 mmol, 67%) as a pale yellow crystalline powder.

**$^1$H NMR (400 MHz, CDCl$_3$):** $\delta$ = 8.09-8.03 (m, 4H, ortho-PPh$_2$), 7.68-7.63 (m, 1H, H$_D$), 7.01-6.94 (m, 7H, PPh$_2$ and H$_C$), 2.69 (s, 3H, H$_G$), 0.15 (s, 9H, H$_F$); **$^{31}$P{$^1$H} NMR (162 MHz, CDCl$_3$):** $\delta$ = 249.0 (d, P$_1$, $^2J_{P1-P2}$ = 97.1 Hz), 34.4 (d, P$_2$, $^2J_{P1-P2}$ = 97.1 Hz, $^{77}$Se satellites (7.63% abundant): dd, $^1J_{Se-P2}$ ≈ 749.1 Hz); **HRMS (ASAP/QTof):** m/z: calcd. for C$_{21}$H$_{24}$P$_2$SeSi: 447.0367 [M+H]$^+$, found: 447.0369; **elemental analysis:** anal. calcd. for C$_{21}$H$_{24}$P$_2$SeSi: C 56.63, H 5.43, found: C 56.89, H 5.37.

*4.5. Preparation of 2-Diphenylphosphineoxide-3-methyl-6-trimethylsilylphosphinine (**10**)*

A solution of 2-diphenylphosphenyl-3-methyl-6-trimethylsilylphosphinine (**6,** 10 mg) in CDCl$_3$ (0.7 cm$^3$) was left under atmospheric oxygen and moisture in an NMR tube. The

reaction progress was monitored using multinuclear NMR spectroscopy and the product by mass spectrometry.

**$^1$H NMR (400 MHz, CDCl$_3$, 298 K)** $\delta$ = 7.74 (m, 1H, $H_D$), 7.65-7.39 (m, 11H, PPh$_2$ + $H_C$), 2.54 (s, 3H, $H_G$), 0.20 (s, 9H, $H_F$); **$^{31}$P{$^1$H} NMR (162 MHz, CDCl$_3$, 298 K)** $\delta$ = 256.2 (d, $P_1$, $^2J_{\text{P1-P2}}$ = 103 Hz), 32.1 (d, $P_2$, $^2J_{\text{P2-P1}}$ = 103 Hz); **$^{13}$C{$^1$H} NMR (100 MHz, CDCl$_3$, 298 K)** (a few signals too weak to be identified) $\delta$ = 138.3 (dd), 130.5-129.8 (multiple signals, phosphinine + PPh$_2$), 126.4 (m), −2.1 (d, SiMe$_3$); **HRMS (ASAP/TOF):** calcd. for C$_{21}$H$_{25}$OSiP$_2$: 383.1150 [M+H]$^+$, found: 383.1144 m/z.

**Author Contributions:** Conceptualisation, S.M.M.; methodology, P.A.C., R.J.N. and S.M.M.; validation, P.A.C., B.G. and S.M.M.; formal analysis, all authors.; investigation, all authors.; data curation, S.M.M.; writing—original draft preparation, S.M.M. and R.W.; writing—review and editing, all authors; visualisation, S.M.M.; supervision, S.M.M.; project administration, S.M.M.; funding acquisition, S.M.M. All authors have read and agreed to the published version of the manuscript.

**Funding:** This research was funded by the Leverhulme Trust (RPG-2016-338 for supporting P.A.C.) and the EPSRC (DTP studentship to R.J.N.).

**Data Availability Statement:** The crystallographic data in this study are available in the Cambridge Structural Database, deposition numbers CCDC 2127803 (**2**), 2127804 (**8**) and 2127805 (**9**).

**Acknowledgments:** We thank the National Mass Spectrometry Facility at Swansea University for help with the mass spectrometry analysis. We thank Brian Hutton (Heriot-Watt University) for help with the elemental analysis and Georgina Rosair (Heriot-Watt University) for collecting the X-ray diffraction data.

**Conflicts of Interest:** The authors declare no conflict of interest.

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
