# Peer review of "Reactivity Studies of Phosphinines: The Selenation of Diphenyl-Phosphine Substituents and Formation of a Chelating Bis(Phosphinine) Palladium(II) Complex"

_inorganics, doi:10.3390/inorganics10020017_

Round 1
Reviewer 1 Report
The manuscript by S. M. Mansell and co-worker is about reactivity studies of diphenylphosphino-substituted phosphinines. In particular, selenation reactions on the Ph2P-group(s) are studied, with respect to the preparation of Pd(II) complexes.
The Mansell group has recently published very interesting results, particularly on the coordination chemistry of substituted phosphinines and their application in catalytic reactions. They report now on the reaction of Ph2P-substituted phosphinines towards selenium (as well as H2O/O2). These reacitons are important, as they can give valuable information on the electronic properties of trivalent phosphorus species via NMR spectroscopy. A series of substituted phosphinines was prepared making use of the 1,3,2-diazaphosphinine route. Interestingly, they authors could isolate and structurally characterize compound 2 (a chlordiazaphosphacycle), which was postulated before to be an intermediate in the conversion of the titanocene-precursor towards the diazaphosphinine. This is a very intriguing result! The selenation reaction of the Ph2P-substituent(s) revealed, that the phosphinine substituent behaves pretty much like a phenyl ring. Finally, the authors report on the sytnthesis of Pd(II) complex, containing a chelating diphosphinine.
The here presented results are very nice and important and contribution to organophosphorus chemistry. The experiments are well conducted and the interesting results are presented in a very concise manner. In my opinion, the manuscript is suitable for publication in Inorganics.
Minor remark: the schematic structure of complex 7 in Scheme 1: even though a distortion is expected, I would rather avoid to use the perspective arrows (P-Pd), as the Pd(II) complex should be square planar.
Author Response
Reviewer 1:
Minor remark: the schematic structure of complex 7 in Scheme 1: even though a distortion is expected, I would rather avoid to use the perspective arrows (P-Pd), as the Pd(II) complex should be square planar.
This has been done.
Reviewer 2 Report
A very good article. Minor reservations are shown in the attachment

Author Response
Reviewer 2:
Line 296:
31P{1H} NMR (162 MHz, CDCl3, 298 K) ? = 234.9 (d, 2JP1-P1 = 100 Hz, P1), 32.9 (dd, 2JP1-P2 = 100 Hz, 5JP2-P3 = 6 Hz, P2), 32.4 (d, 5JP2-P3= 6 Hz, P3).Explain what is 2JP1P1 ;
No JP1- P3 coupling?
This should be P1-P2 and has been corrected. No P1-P3 coupling was observed (it may be small and not obvious compared to the line width).
Lines 313 -314:
31P{1H} NMR (162 MHz, CDCl3): δ = 249.0 (d, P1, 2JP1-P2 = 100.6 Hz), 34.4 (d, P2, 2JP1-P2 = 97.1 Hz),
Why are the coupling constants P1-P2 different for P1 and P2?
Re-inspection shows that both peaks are doublets with 97.1 Hz coupling. We have corrected this.
Reviewer 3 Report
In general, I find this paper suitable for this journal (in terms of subject matter), and the experiments carried out give a solid foundation for the discussion. Therefore, I recommend publication upon some minor revision. The following points should be taken into consideration:
- Typo in line 74 “that that”
- A bold-style 11 should be added to scheme 1 right after “additional product” (because 11 is mentioned in the discussion).
- Line 201: It should read “correct” (rather than “correction”)
- Part 3 “Discussion” should be called “Summary”
- Line 233 “MBRAUN”
- Lines 241-243: It should be mentioned (in a more detailed fashion) which signal has been used for referencing. For 1H, a protio residual solvent signal must have been used. CDCl3 cannot produce a 1H NMR signal, it must have been residual CHCl3. In 13C, both CDCl3 and residual CHCl3 could have been used. It must be pointed out, which signal has been used. In case of the 13C CDCl3 signal, it is the actual solvent signal (not the residual solvent signal).
- Please check the correct NMR shift values reported (for all compounds). For example, in lines 241/242 it reads that 1H has been referenced to solvent at 7.24 ppm, and line 266 gives a 1H NMR shift at 6.30 ppm, but Figure SI1 (1H NMR of this compound) shows the shift of 6.30 ppm of this signal within a spectrum in which the CHCl3 signal is sitting at 7.29 ppm, hence, it has not been reported relative to the residual solvent signal at 7.24 ppm.
Author Response
Reviewer 3:
- Typo in line 74 “that that”
Corrected
- A bold-style 11 should be added to scheme 1 right after “additional product” (because 11 is mentioned in the discussion).
Done
- Line 201: It should read “correct” (rather than “correction”)
Done
- Part 3 “Discussion” should be called “Summary”
Done
- Line 233 “MBRAUN”
Done
- Lines 241-243: It should be mentioned (in a more detailed fashion) which signal has been used for referencing. For 1H, a protio residual solvent signal must have been used. CDCl3 cannot produce a 1H NMR signal, it must have been residual CHCl3. In 13C, both CDCl3 and residual CHCl3 could have been used. It must be pointed out, which signal has been used. In case of the 13C CDCl3 signal, it is the actual solvent signal (not the residual solvent signal).
This has been done:
1H NMR spectra were recorded at 400 MHz and referenced to the residual protio solvent peak (7.24 for CHCl3 in CDCl3 and 7.16 for C6D5H in C6D6). 13C{1H} NMR spectra were recorded at 101 MHz and referenced to the solvent peak (77.23 ppm for CDCl3 and 128.39 ppm for C6D6).
- Please check the correct NMR shift values reported (for all compounds). For example, in lines 241/242 it reads that 1H has been referenced to solvent at 7.24 ppm, and line 266 gives a 1H NMR shift at 6.30 ppm, but Figure SI1 (1H NMR of this compound) shows the shift of 6.30 ppm of this signal within a spectrum in which the CHCl3 signal is sitting at 7.29 ppm, hence, it has not been reported relative to the residual solvent signal at 7.24 ppm.
We have reprocessed and correctly referenced all 1H and 13C NMR spectra. These have been added to the ESI and the correct NMR chemical shifts have been used in the experimental section of the paper.
Reviewer 4 Report
The work of Mansell et al. deals with the preparation and structural studies of a few phosphinophosphinins (4, 5 and 6) as well as selenide derivatives 8 and 9. Complexation of 4 with Pd (II) was achieved, but its characterization by X-ray diffraction It was not possible.
Overall, the job is well done and the conclusions are consistent. The references on this type of selenide derivatives are scarce and this study deserves to be published in Inorganics.
Author Response
Reviewer 4: No corrections required.
Reviewer 5 Report
In the paper, Dr Stephen Mansell study the reactivity of several phosphinines.
The introduction is well documented offering a good overview of their development and applications. The comparison between phosphine oxidation (O, S, Se) and phosphinine allow a good understanding of the results obtained later and their applications as ligands justify the chosen multidentate ligands. Some new results such as the structural characterization of the chlorodiazaphosphacycle 2 is a nice addition to this chemistry together with some new organometallic Pd complex.
Overall, this is an interesting study, well conducted, with good experimental investigations that can be useful for future development of phosphinine chemistry. This paper may thus be accepted in Inorganics, after minor revisions:
#1 Even if molecules 4, 5 and 6 have been prepared in previous articles, a sentence should be added to the text to describe their synthesis, helping the reader to follow the story (between the characterization of 2 and reaction of 4).
#2 Concerning the palladium complex 7, taking in account the modest yield obtained, it would be nice to precise (either in experimental section or in the article) if the coordination is selective (analysis of the crude sample by 31P). Monitoring the reaction by 31P could help, avoiding treatment whereas some free ligand is remaining in solution. The optimization of the time could help to improve the yield. Do the reaction conditions lead to degradation of PdCl2COD?
#3 For selenation reaction of 5 (and also 6), a full conversion is mentioned in the article, but the isolated yield is only 13%. Why? Could performing the reaction under pressure like in the case of 6 could help? Is there selectivity or extraction problems? The authors should comment this.
#4 On p7 l 223, based on similar 1JP-Se coupling, the authors claim that the phosphinine rings acts as a phenyl ring which is true. However, the last part of the sentence :“and thus the disphenylphosphinine donors resemble to PPh3 in donor properties” is for me misleading and should be removed. It’s quite negative in regard of the major topic of the article, the phosphinines!
Author Response
Reviewer 5:
#1 Even if molecules 4, 5 and 6 have been prepared in previous articles, a sentence should be added to the text to describe their synthesis, helping the reader to follow the story (between the characterization of 2 and reaction of 4).
We have added the following:
They were synthesised by the reaction of the diazaphosphinine 3 with alkynes in two sequential steps due to the slower reaction of the intermediate azaphosphinines with alkynes. This allowed two different alkynes to be used in the formation of 4 and 6.
#2 Concerning the palladium complex 7, taking in account the modest yield obtained, it would be nice to precise (either in experimental section or in the article) if the coordination is selective (analysis of the crude sample by 31P). Monitoring the reaction by 31P could help, avoiding treatment whereas some free ligand is remaining in solution. The optimization of the time could help to improve the yield. Do the reaction conditions lead to degradation of PdCl2COD?
Re-checking through our NMR spectra shows that the reaction is selective, and we have added this to the paper:
The reaction is selective, with only one product observed. We have not found PdCl2COD to be degraded.
#3 For selenation reaction of 5 (and also 6), a full conversion is mentioned in the article, but the isolated yield is only 13%. Why? Could performing the reaction under pressure like in the case of 6 could help? Is there selectivity or extraction problems? The authors should comment this.
We have re-investigated this reaction several times and found the diselenide to be poorly soluble in toluene when the solid product is extracted, however, it is soluble in the toluene reaction mixture. We have adapted the procedure to use THF as the extracting solvent and have improved the yield (45%).
#4 On p7 l 223, based on similar 1JP-Se coupling, the authors claim that the phosphinine rings acts as a phenyl ring which is true. However, the last part of the sentence :“and thus the disphenylphosphinine donors resemble to PPh3 in donor properties” is for me misleading and should be removed. It’s quite negative in regard of the major topic of the article, the phosphinines!
We have removed this from the paper.